# Comparable respiratory activity in attached and suspended human fibroblasts

**Lucie Zdrazilova**[1]*, **Hana Hansikova**[1], **Erich Gnaiger**[2]*

**1** Department of Pediatrics and Inherited Metabolic Disorders, First Faculty of Medicine, Charles University and General Hospital in Prague, Prague, Czechia, **2** Oroboros Instruments, Innsbruck, Austria

* Lucie.Zdrazilova@lf1.cuni.cz (LZ); erich.gnaiger@oroboros.at (EG)

**Data Availability Statement:** Original files are available Open Access at Zenodo repository: 10.5281/zenodo.5518059

**Funding:** This work was partially funded by the European Union's Horizon 2020 research and

## Abstract

Measurement of oxygen consumption of cultured cells is widely used for diagnosis of mitochondrial diseases, drug testing, biotechnology, and toxicology. Fibroblasts are cultured in monolayers, but physiological measurements are carried out in suspended or attached cells. We address the question whether respiration differs in attached versus suspended cells using multiwell respirometry (Agilent Seahorse XF24) and high-resolution respirometry (Oroboros O2k), respectively. Respiration of human dermal fibroblasts measured in culture medium was baseline-corrected for residual oxygen consumption and expressed as oxygen flow per cell. No differences were observed between attached and suspended cells in ROUTINE respiration of living cells and LEAK respiration obtained after inhibition of ATP synthase by oligomycin. The electron transfer capacity was higher in the O2k than in the XF24. This could be explained by a limitation to two uncoupler titrations in the XF24 which led to an underestimation compared to multiple titration steps in the O2k. A quantitative evaluation of respiration measured via different platforms revealed that short-term suspension of fibroblasts did not affect respiratory activity and coupling control. Evaluation of results obtained by different platforms provides a test for reproducibility beyond repeatability. Repeatability and reproducibility are required for building a validated respirometric database.

## Introduction

Studies of cells attached in a monolayer or suspended in the medium have wide-ranging applications and implications. These include metabolic profiling [1], substrate diffusion [2], cell morphology and rheology [3], macrophage adherence [2], suspension culture as mimetic of circulating tumor cells [4], metastatic potential [4], therapeutic cell reimplantation, and cell culture biotechnology and pharmacology [5].

Fibroblast cell lines are established models routinely applied in studies of mitochondrial diseases [6–9]. These cells can be investigated in culture either attached to the surface of an experimental chamber or in suspension after detachment.

The structure of cells growing in culture changes from the attached to the suspended state. After trypsinization, fibroblasts undergo membrane reorganization and attain a spherical

innovation program under grant agreement No. 859770, NextGen-O2k project (EG), Institutional projects of Charles University GAUK110119 and SVV–UK 260367 (LZ) and by the Ministry of Health of the Czech Republic NV19-07-00149 (HH). Contribution to COST Action CA15203 MitoEAGLE with financial support of Short-Term Scientific missions (LZ).

**Competing interests:** We have read the journal's policy and the authors of this manuscript have the following competing interests: EG is founder and CEO of Oroboros Instruments, Innsbruck, Austria. The sV chambers were a loan provided by Oroboros Instruments. The authors declare not to have any other competing interests. This does not alter our adherence to PLOS ONE policies on sharing data and materials.

**Abbreviations:** amol, attomole ($10^{-18}$ moles); ace, attached cells; DMEM, Dulbecco's modified Eagle's medium; $E$ and $E'_{tot}$, ET capacity per cell, *Rox*-corrected and total [amol·s$^{-1}$·x$^{-1}$]; $(E\text{-}L)_{50}$, net ET capacity when coupling efficiency is 50% in the hyperbolic dyscapacity model; $(E\text{-}L)_{50} = L$; $E_{50} = 2 \cdot L$; EDTA, Ethylenediaminetetraacetic acid; ETS, electron transfer system; FCCP, Carbonyl cyanide 4-(trifluoromethoxy)phenylhydrazone; HRR, high-resolution respirometry; $I_{O2}$, O$_2$ flow per cell count; j, coupling control efficiency $(E\text{-}L)/E$; $j_{max}$, maximum coupling control efficiency, by definition $j_{max} \cong 1$; $J_{O2}$, O$_2$ flux per chamber volume; $L$ and $L'_{tot}$, LEAK respiration per cell, *Rox*-corrected and total [amol·s$^{-1}$·x$^{-1}$]; M, mega ($10^6$); M, molar (mol·L$^{-1}$); mt, mitochondria(l); n, number of technical repeats or total number of measurements; N, number of independent replica; $N_{ce}$, cell count, number of cells [x]; OCR, oxygen consumption rate; p, pico ($10^{-12}$); $R$ and $R'_{tot}$, ROUTINE respiration per cell, *Rox*-corrected and total [amol·s$^{-1}$·x$^{-1}$]; ROX, residual oxygen consumption state; Rox, residual oxygen consumption (per cell [amol·s$^{-1}$·x$^{-1}$]); sce, suspendeded cells; SUIT, substrate-uncoupler-inhibitor titration; TE, trypsin 0.05% w/V with EDTA 0.02%, w/V; U, uncoupler; x, elementary unit.

shape with a so-called blebbed surface morphology to prevent membrane loss by providing transient membrane storage [3]. In rabbit lung macrophages, transport of lysine and adenosine across the plasma membrane is faster in suspended cells compared to adherent ones [2]. Moreover, mouse macrophages oxidize glucose six times faster when in suspension than in monolayers [1]. Suspending anchorage-dependent fibroblasts results in an abrupt drop of mRNA production, while protein synthesis declines slowly but extensively and its recovery requires surface contact [10, 11]. Taken together, these observations raise the physiological question whether cell respiration differs in suspended versus attached fibroblasts. Stimulatory or suppressive effects may be exerted on aerobic ATP demand and consequently respiration may be regulated differently in suspended and attached states.

The first aim of the present study was the evaluation of respiration in attached compared to suspended fibroblasts. The Seahorse XF Analyzer (Agilent, US) is designed for studying respiration of attached cells (ace), whereas the Oroboros O2k (Oroboros Instruments, Austria) is optimized for high-resolution respirometry with suspended cells (sce). Therefore, a platform comparison was required as the second aim of our study. We applied comparable and standardized protocols for monitoring respiratory activity in living cells under near-physiological conditions. Specific titration steps in the protocol disrupt the physiological control of respiration, which then allows for instrumental comparison of performance of the two platforms. Respiratory activity was normalized per cell and expressed in identical SI units for a quantitative comparison of the data. The experimental period of suspended cells was limited to less than one hour. The respiratory activity of suspended and attached cells was not distinguishable during such short time intervals. Comparative studies provide the basis for extending databases of cellular respiration by including critically evaluated results obtained with different platforms.

# Materials and methods

## Reagents

DMEM5030, Carbonyl cyanide 4-(trifluoromethoxy)phenylhydrazone (FCCP, C2920-10MG), oligomycin(O4876-5MG), antimycin A (A8674-25MG), rotenone (R8875-1G), and glutamine (G7513-100mL) were purchased from Sigma Aldrich. Trypsin- ethylenediaminetetraacetic acid EDTA 10x (XC-T1717/100) and antibiotics-antimycotics 100X (XC-A4110/100) were obtained from Biosera, Dulbecco's modified Eagle's medium (DMEM, P04-04510) from Pan Biotech, fetal bovine serum (SV30160.03) from HyClone, and XF Calibrant Solution 100840–000 from Agilent.

## The instruments

The Seahorse XF Analyzer (Agilent, Santa Clara, US) provides multi-well plate analysis of two processes in real time: oxygen consumption rate (OCR) as an indicator of cell respiration, and extracellular acidification rate largely dependent on glycolytic processes. Cellular O$_2$ consumption causes changes in the concentration of dissolved dioxygen O$_2$ in so-called 'transient microchambers'. O$_2$ pressure is measured by solid-state fluorescent probes and converted to O$_2$ concentration. Every measurement step takes 5–8 min during which O$_2$ concentration is measured every few seconds, providing data for OCR calculation. Then the probes are lifted, and the larger volume of medium is mixed to restore O$_2$ levels to baseline conditions. Chemical compounds are injected pneumatically, limited to four sequential injections per well [12].

The Oroboros O2k (Oroboros Instruments, Innsbruck, Austria) is a two-chamber high-resolution respirometer used in cell and mitochondrial research to measure respiration in mitochondrial preparations and living cells. The O2k supports multi-sensor modules and

**Table 1. Comparative specifications of Seahorse XF24 and Oroboros O2k applied in the present coupling control protocol.**

| | Seahorse XF24 | Oroboros O2k |
|---|---|---|
| wells or chambers per instrument | 24 semiclosed wells | two diffusion-tight chambers |
| instrumental background and experiments with cells | four separated wells without cells and 20 wells with cells in parallel | two identical chambers serially without and with cells |
| detection mode | optical $O_2$ sensor with fluorophores | electrochemical polarographic oxygen sensor |
| temperature control | 37˚C (stability ±0.1˚C in a tray) | 37˚C (stability ±0.002˚C in a copper block) |
| limit of detection of oxygen flux | not specified | ±1 pmol $O_2 \cdot s^{-1} \cdot mL^{-1}$ |
| cell conditions | attached cells | cell suspension |
| required cell amount | 35 000 | 375 000 |
| sample volume [μL] | 450 | 540 |
| closed chamber volume | 7 μL when sensor probes are inserted | 500 μL when stoppers are inserted |
| titrations | automatic | manual |
| number of possible titrations | 4 | technically unlimited |
| experimental duration [min] | 120 | 30 to 40 |

measurement is performed in the experimental chamber, where suspended cells or mitochondrial preparations are continuously mixed by a stirrer at 750 rotations per minute. $O_2$ consumption in nearly diffusion-tight closed chambers is calculated in real time from $O_2$ partial pressure measured by polarographic oxygen sensors. The O2k provides the option of practically unlimited titrations and, therefore, the possibility to apply various Substrate-Uncoupler-Inhibitor-Titration (SUIT) protocols designed to address specific research questions [13, 14]. Table 1 summarizes the comparison between the two instruments.

## Cell culture

Two human dermal fibroblast cell lines were purchased, HDF 1 (Primary Dermal Fibroblast Normal; Human, Neonatal HDFn, PCS_201_010, ATCC, NHDF-Neo), and HDF 2 (Human Dermal Fibroblasts, Neonatal, CC-2509, Lonza). A human dermal fibroblast cell line (HDF 3) was derived from a disease-free control at age of 5 months upon verbal informed parentals' consent obtained in General Hospital in Prague with ethics committee approval N° 92/18 (18.10.2018) for the project GAUK (Grant Agency of Charles University) 110119. All cell lines were cultured in Dulbecco's modified Eagle medium (DMEM, Pan Biotech) with 25 mM glucose, 10% fetal bovine serum and 1% antibiotics-antimycotics 100X at 37˚C under 5% $CO_2$ atmosphere. Fibroblast cultures at passage number 13 to 15 were grown to approximately 80% confluence. Suspended cells were counted by a Handheld Automated Cell Counter (Millipore). Cells were harvested by incubation in trypsin 0.05% w/V with EDTA 0.02%, w/V for 5 min at 37˚C, washed, and centrifuged at 300 g (5 min, 24˚C).

## Sample preparation and respirometry

DMEM5030 was the basis of DMEM respiration medium with addition of 3.9 mM glutamate, 5 mM glucose and 2 mM pyruvate, adjusted to pH 7.4 at 37˚C. DMEM respiration medium was freshly prepared on the day of use.

Mitochondrial respiration was measured in the Department of Pediatrics and Inherited Metabolic Disorders, Prague. The Agilent Seahorse XF Analyzer (XF24) according to [15] with slight modifications. The day before measurement, cells were harvested after trypsinization, resuspended in DMEM culture medium, counted, and 35 000 cells were seeded on 20 wells of a 24-well plate for overnight incubation. 4 wells were used as blanks without cells. The plates were incubated overnight in a 5% $CO_2$ atmosphere at 37˚C. In parallel, the Sensor Cartridge

was hydrated in wells filled with Seahorse XF Calibrant Solution (Agilent) by incubation without $CO_2$ over-night prior to use. The XF24 was switched on during the day before experiments to equilibrate at 37˚C. On the following day, wells were washed twice with 1 mL DMEM respiration medium. 450 μL DMEM respiration medium was added to each well and incubated without $CO_2$ for one hour at 37˚C. In the meantime, 50 μL oligomycin (stock 20 μM; experimental concentration 2 μM) was added to cartridge port A, 55 μL FCCP (stock 5 μM) to port B, 61 μL FCCP (stock 2 μM) to port C (experimental concentrations 0.5 and 0.7 μM, respectively), and 67 μL rotenone (stock 20 μM) with antimycin A (stock 10 μM) to port D (experimental concentrations 2 and 1 μM). Sensor cartridges were incubated without $CO_2$ for 30–40 min at 37˚C, transferred to the XF24 for equilibration and calibration in Seahorse XF Calibrant Solution. Then the calibration well plate was exchanged for the cell plate. Before starting the respiratory protocol with living cells, wells were mixed for 3 min and left idle for 2 min. Respiratory flux was measured in each state three times for 3 min. The transient microchamber for measurement had a volume of approximately 7 μL, with sensors positioned 200 μm above the well bottoms.

Air calibration in the Oroboros O2k was performed daily before measurement [16]. The $O_2$ partial pressure for air calibration is calculated (DatLab software) for air saturated 100% with water vapor at experimental temperature and local barometric pressure recorded real-time by the electronic pressure transducer of the O2k. The $O_2$ solubility of DMEM respiration medium was assumed to be 92% relative to the $O_2$ solubility of pure water at 37˚C [17]. Calibrations at zero $O_2$ concentration were performed before the experimental series. Instrumental $O_2$ background flux [18] was measured each day before the experiment with cells in the range of $O_2$ concentration from air saturation at 190 μM to 100 μM.

Experimental oxygen concentrations near air saturation (190 μM to 140 μM; Fig 1B and 1D) are much higher than extracellular oxygen concentrations in various tissues in vivo. Respiration of living cells, however, is independent of oxygen pressure from air saturation to tissue-level oxygen concentration, the latter ranging between 50 and 10 μM [19]. Therefore, even if high experimental oxygen concentrations are not physiological, respiratory measurements are not affected. This is in direct contrast to cellular hydrogen peroxide production, which increases with oxygen concentration over the entire experimental oxygen concentration range [20].

## Respiratory protocol

Comparable respiratory protocols were used with the Agilent XF24 and Oroboros O2k using the same DMEM respiration medium (Fig 1). Every cell line was measured on the same day in parallel, splitting the cells from one culture flask into two O2k -chambers and 20 wells of the XF24. In the coupling control protocol for living cells (SUIT-003), four respiratory states are distinguished [21]. We adhere to the platform-independent terminology of the MitoEAGLE consensus paper [22] and add the terms widely used in conjunction with the XF24 or XF96 [15] in parentheses (Table 2).

First, ROUTINE respiration $R$ ('basal respiration') was measured in attached or suspended cells. The ROUTINE state is a physiological state, in which respiration is controlled by cellular energy demand. Next, the ATP synthase inhibitor oligomycin was added to induce LEAK respiration $L$ ('proton leak'). In the non-phosphorylating LEAK state, a low rate of respiration is maintained mainly to compensate for the proton leak at a high protonmotive force. Afterwards, the uncoupler FCCP was titrated in at least two steps to obtain a maximum rate of $O_2$ consumption reflecting the electron transfer capacity $E$ ('maximal respiration') in the noncoupled state. Finally, antimycin A and rotenone were added. These inhibitors of Complexes CI and CIII, respectively, inhibit mitochondrial electron transfer and thus induce residual

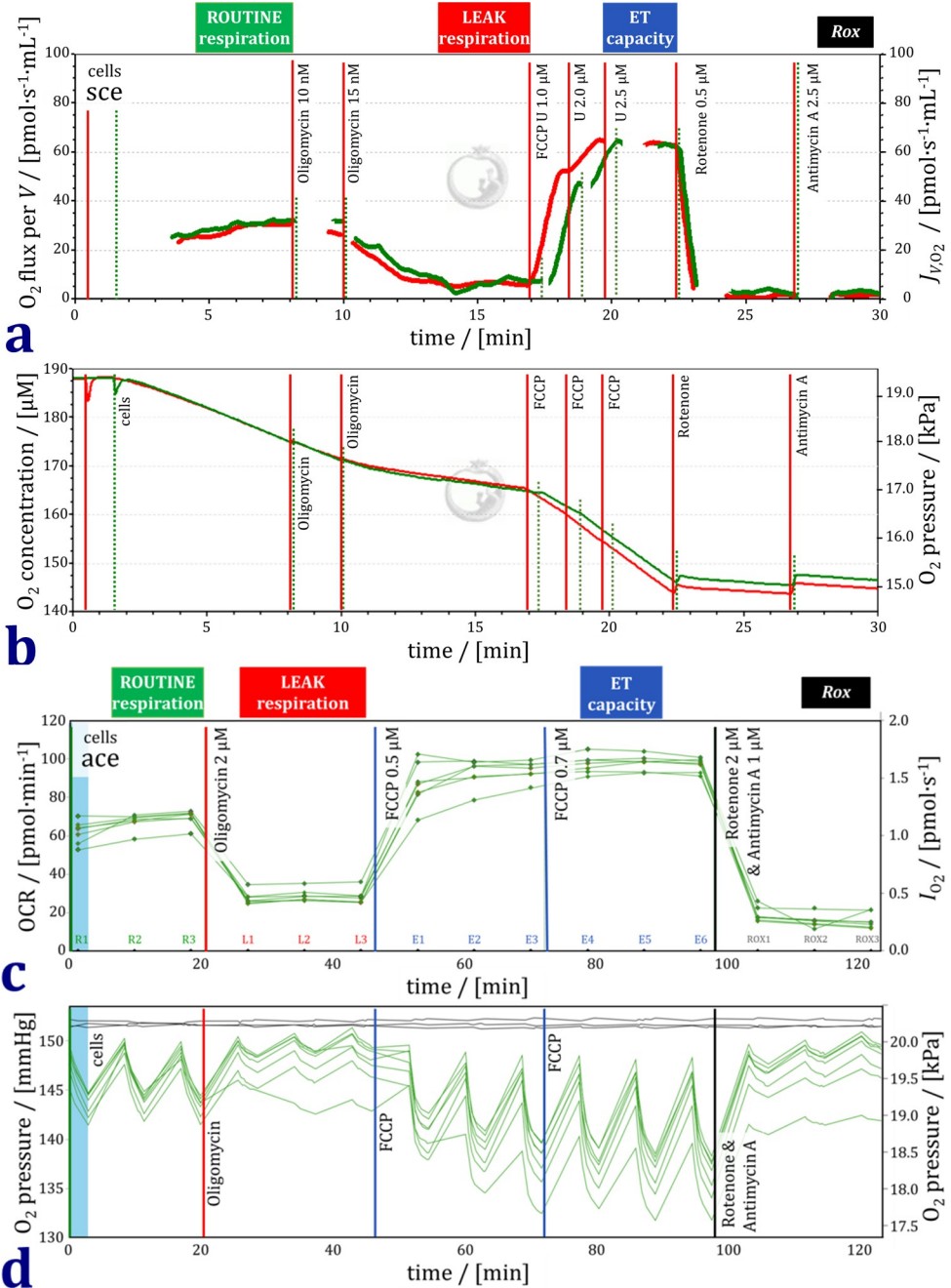

**Fig 1. Representative traces of respiration and O₂ concentration or O₂ pressure in the protocol with living human dermal fibroblasts (HDF 1).** (a) $O_2$ flux per volume in two chambers measured simultaneously in the Oroboros O2k. (b) $O_2$ concentration and partial $O_2$ pressure corresponding to the traces in panel a. (c) $O_2$ consumption rate (OCR) measured simultaneously in six wells of one plate of the XF24. Three time intervals of respiration per respiratory state (R1, R2, R3; L1, L2, L3; etc.). (d) $O_2$ partial pressure corresponding to the traces in panel c.

oxygen consumption *Rox* ('nonmitochondrial respiration') due to oxidative side reactions. Importantly, the plasma membrane is permeable for these inhibitors and the uncoupler which, therefore, can be applied in living cells. The concentrations applied in the XF24 and O2k adhered closely to the respective manuals (Table 3; Fig 1).

**Table 2. Harmonization of terminology on respiratory states.**

| | MitoEAGLE Task Force 2020 | Seahorse XF Analyzer | Definition |
|---|---|---|---|
| $R$ | ROUTINE respiration | basal respiration | physiological respiration controlled by cellular energy demand, energy turnover and the degree of coupling to phosphorylation |
| $L$ | LEAK respiration | proton LEAK | non-phosphorylating state, respiration maintained mainly to compensate for the proton leak at a high chemiosmotic potential |
| $E$ | electron transfer capacity | maximal respiration | oxygen consumption with a short circuit of the $H^+$ cycle across the mitochondrial inner membrane stimulating maximum O2 flux |
| $Rox$ | residual oxygen consumption | nonmitochondrial respiration | respiration due to oxidative side reactions in the ROX-state after Complex I and III inhibition |

Comparison of platform-independent MitoEAGLE terms [22] and terms frequently used in the context of Seahorse XF Analyzer applications [23]. $R$, $L$, and $E$ are baseline-corrected for $Rox$.

## Data analysis

Measurements were normalized for the cell count. Traces were analyzed in DatLab 7 software (Oroboros Instruments) where each O2k -chamber contained 375 000 cells. $O_2$ flow is expressed per cell [amol·s$^{-1}$·x$^{-1}$] equivalent to [pmol·s$^{-1}$·(10$^6$ x)$^{-1}$]. The Wave 2 software (Agilent) presents results in units [pmol·min$^{-1}$] in a well which contained 35 000 cells. Data were converted to the same units [amol·s$^{-1}$·x$^{-1}$], where x represents the unit cell [14, 24]. In regressions between respiration for different states, variables in $X$ and $Y$ have comparable errors of measurement. To minimize the residuals of both variables, $Y$ and $X$, slopes $b_Y$ and $\beta_X$ and intercepts $a_Y$ and $\alpha_X$ are calculated for the $Y/X$ and $X/Y$ inverted linear regressions, respectively. The mean slope $\bar{b} = (b_Y + b_X)/2$ and mean intercept $\bar{a} = (a_Y + a_X)/2$ are used, where $b_X = 1/\beta_X$ and $a_X = -\alpha_X/\beta_X$ [25]. Further statistical evaluation was performed using Prism (GraphPad Software, California, USA). $Rox$-corrected rates were symmetrically distributed, and logarithmic transformation was not required. One outlier was removed in the XF24 with negative LEAK respiration irrespective or $Rox$-correction. There were no outliers in the O2k.

## Results

### Respiration normalized for cell count

Respiratory rates measured in the XF24 (ace) and O2k (sce) are expressed per cell in units for $O_2$ flow [amol·s$^{-1}$·x$^{-1}$] and shown on identical scales in Fig 2A and 2B. The ranking of respiratory capacities of the three cell lines followed different patterns in the XF24 and O2k, which was taken as an argument for pooling all results in the scatter plots. The variability within cell lines was greater in the XF24 ($n = 20$) than the O2k ($n = 4$). Averaging five wells of the XF24 to obtain $n = 4$ per cell line did not reduce the coefficient of variation (SD/average), which was

**Table 3. Recommended and applied experimental concentrations of inhibitors and uncoupler [14; 23].**

| | Compound | Gnaiger (2020) | O2k applied | Agilent (2019) | XF24 applied |
|---|---|---|---|---|---|
| $L$ | Oligomycin | 5–10 nM titration steps | 10–15 nM | 0.5–2.5 µM | 2 µM |
| $E$ | FCCP | 0.5 µM titration steps CCCP | 2–2.5 µM | 0.125–2.0 µM | 0.7 µM |
| | | | FCCP | FCCP | FCCP |
| $Rox$ | Rotenone | 0.5 µM | 0.5 µM | 0.5 µM | 2 µM |
| | Antimycin A | 2.5 µM | 2.5 µM | 0.5 µM | 1 µM |

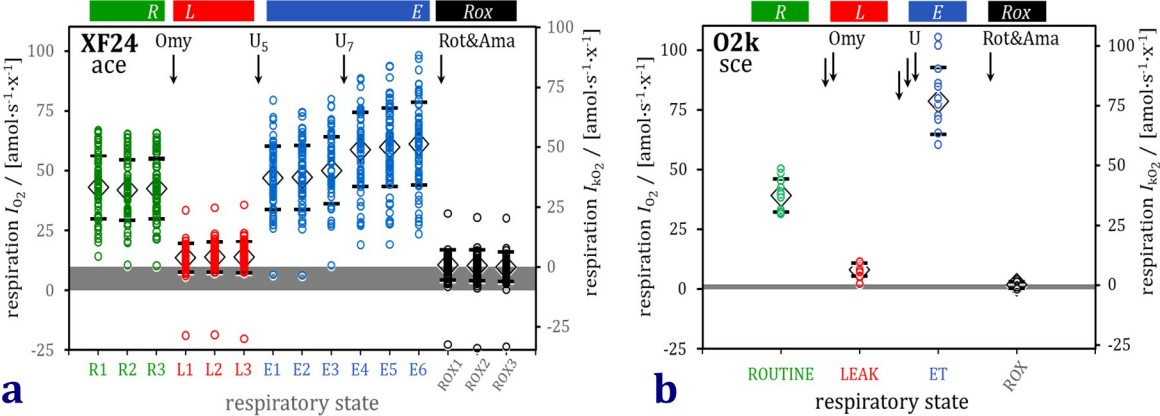

**Fig 2.** Sequence of respiratory states in the coupling control protocol applied in the XF24 for ace (a) and O2k for sce (b). Circles: individual wells or chambers; diamonds and bars: median ± SD. ROUTINE respiration $R$; LEAK respiration $L$; electron transfer capacity $E$; residual oxygen consumption $Rox$.

0.29 and 0.26 for $R$ and $E$ ($n = 20$) to 0.27 and 0.27 ($n = 4$, each for 5 pooled wells), compared to the coefficient of variation of 0.10 and 0.10 for $R$ and $E$ in the O2k (Table 4A).

Rates of $O_2$ consumption are expected to be identical in adherent and suspended cells in the LEAK, ET, and ROX states when physiological control of cellular energy demand is eliminated. This is achieved by inhibition of mitochondrial electron transfer using rotenone and antimycin A. These block the Complexes I and III, respectively, and thus induce the state of

**Table 4. Respiration of three human fibroblast cell lines attached (ace; XF24) or suspended (sce; O2k).**

| **a** | | Cell | Number of | ROUTINE R | LEAK L | ET capacity E | *Rox* |
|---|---|---|---|---|---|---|---|
| | | line | repeats n or groups | [amol·s-1·x-1] | [amol·s-1·x-1] | [amol·s-1·x-1] | *[amol·s-1·x-1]* |
| ace$_1$ | | | $n = 20$ | 33 ± 5 | 5 ± 2 | 51 ± 7 | 8 ± 8 |
| ace$_2$ | | | $n = 20$ | 39 ± 10 | 6 ± 2 | 60 ± 14 | 11 ± 6 |
| ace$_3$ | | | $n = 19$ | 30 ± 13 | 5 ± 2 | 44 ± 18 | 7 ± 4 |
| sce$_1$ | | | $n = 4$ | 45 ± 4 | 7 ± 1 | 94 ± 9 | 2 ± 1 |
| sce$_2$ | | | $n = 4$ | 35 ± 4 | 6 ± 3 | 75 ± 7 | 1 ± 2 |
| sce$_3$ | | | $n = 4$ | 31 ± 3 | 3 ± 1 | 69 ± 7 | 2 ± 2 |
| ace$_{mean}$ | | | $N = 3$ | 34 ± 4 | 5 ± 1 | 52 ± 8 | 9 ± 2 |
| sce$_{mean}$ | | | $N = 3$ | 37 ± 7 | 6 ± 2 | 79 ± 13 | 2 ± 1 |
| **b** | | Cell | Number of | control ratio | net *R/E* ratio | net E ratio | ROX |
| | | line | replica n or groups N | R/E | (R-L) /E | (E-L)/E | *Rox*/E'$_{tot}$ |
| ace$_1$ | | | $n = 20$ | 0.64 ± 0.07 | 0.56 ± 0.06 | 0.91 ± 0.06 | 0.14 ± 0.27 |
| ace$_2$ | | | $n = 20$ | 0.61 ± 0.10 | 0.56 ± 0.11 | 0.89 ± 0.04 | 0.15 ± 0.08 |
| ace$_3$ | | | $n = 19$ | 0.71 ± 0.07 | 0.60 ± 0.08 | 0.92 ± 0.07 | 0.15 ± 0.06 |
| sce$_1$ | | | $n = 4$ | 0.46 ± 0.04 | 0.38 ± 0.04 | 0.92 ± 0.00 | 0.02 ± 0.01 |
| sce$_2$ | | | $n = 4$ | 0.49 ± 0.06 | 0.39 ± 0.03 | 0.91 ± 0.04 | 0.01 ± 0.02 |
| sce$_3$ | | | $n = 4$ | 0.45 ± 0.04 | 0.41 ± 0.02 | 0.96 ± 0.02 | 0.03 ± 0.02 |
| ace$_{mean}$ | | | $N = 3$ | 0.61 ± 0.05 | 27.4 ± 4.7 | 0.90 ± 0.03 | 0.16 ± 0.01 |
| sce$_{mean}$ | | | $N = 3$ | 0.47 ± 0.02 | 31.2 ± 5.9 | 0.93 ± 0.03 | 0.02 ± 0.01 |

Median ± SD. **(a)** $R$, $L$ and $E$ corrected for residual oxygen consumption $Rox$. $n$ are technical repeats. **(b)** Flux control ratios and flux control efficiencies normalized for ET capacity as an internal normalization to express respiration independent of cell count. $Rox$ was normalized for total $O_2$ consumption $E'_{tot}$ in the noncoupled state. $E'_{tot}$ was determined immediately before inhibition by antimycin A and is the electron transfer capacity without correction for $Rox$.

residual oxygen consumption ROX. Similarly, inhibition of ATP synthase by oligomycin induces LEAK respiration $L$ in a state that is not controlled by cellular ATP turnover. ROX and LEAK are, therefore, states of minimum $O_2$ consumption, providing a comparison of instrumental resolution of the two instrument types, independent of using attached cells (ace) in the XF24 and suspended cells (sce) in the O2k. Residual oxygen consumption $Rox$ and total LEAK respiration $L'_{tot}$ not corrected for $Rox$ were higher with a larger scatter in the XF24 (ace) compared to the O2k (sce) (Fig 3A and 3B). $Rox$-correction of LEAK respiration $L$, however, eliminated the differences observed in the two platforms (Fig 3C). Bioenergetic cluster analysis BCA [25] shows the pairwise correlation between $Rox$ and total LEAK respiration (Fig 3D and 3E).

ROUTINE respiration $R$ is under physiological control of living cells. Therefore, $R$ reflects the possible changes in mitochondrial ATP demand induced by suspending cells that were grown attached in a monolayer. $Rox$-corrected $R$ was similar in attached cells measured in the XF24 and suspended cells measured in the O2k (Fig 3F).

Electron transfer capacity $E$ is supported by physiological substrates in the living cell but is entirely independent of respiratory control by cellular ATP demand. Therefore, $E$ corrected for $Rox$ was expected to be independent of cell physiology and comparable in suspended and attached cells. However, $E$ was lower in ace than sce (Fig 3G). Results for the three cell lines are summarized in Table 4.

## ROUTINE versus LEAK respiration and electron transfer capacity

Internal normalization eliminates any possible bias caused by methodological differences in determining the concentration of cells per experimental chamber volume in wells (attached; ace) and in suspension (sce).

ET capacity is frequently used as a reference and functional mt-marker (Table 4B). The higher $R/E$ flux control ratio in ace, however, does not indicate a lower ROUTINE respiration in sce, which would contradict the results on $O_2$ flow normalized for cell count (Fig 3). BCA illustrates a left shift of the $R/E$ regression line due to lower ET capacity in ace compared to sce, at comparable ROUTINE respiration of suspended and attached cells (Fig 4A). The compensatory increase of $R$ as a function of intrinsic uncoupling detected by increasing $L$ in sce (Fig 4B) conforms to a pattern generally observed in cell respiration [25]. Noise in the ace data prevents resolution, yet the overlap of clusters supports the conclusion based on $O_2$ flow (Fig 3F) that ROUTINE respiration was not different in ace and sce.

Low ET capacities in ace are explained by the high oligomycin concentration applied in the XF24, which may inhibit the ET capacity compared with the minimum oligomycin concentration optimized by titrations in each experiment conducted in the O2k. In addition, the trend of increasing respiration in response to the second and low uncoupler titration suggests that O2 flow would further increase in the actual ET state (Fig 2A).

Declining $E$ at constant $L$ lowers the $E$-$L$ coupling efficiency, $j = (E\text{-}L)/E$, by an ET-linked mechanism. The distinction between ET-linked and uncoupling-linked effects, however, is not apparent from the ratios presented in Table 4B. When $E$ and $L$ are expressed as $O_2$ flow per cell (Table 4A) and vary proportionally due to differences in cell size, mt-density, and noise in the cell count, then coupling efficiency remains constant at variable ET capacity, as seen in sce (Fig 4C). Even when $E$ is underestimated progressively, the drop in $(E\text{-}L)/E$ is small initially, since the decline in $(E\text{-}L)$ is partially compensated for by the decline of $E$. The relationship between coupling efficiency and declining $E$ at constant $L$ is hyperbolic. The XF24 data follow this nonlinear model as a separate cluster overlapping with the O2k data at high coupling efficiency (Fig 4C).

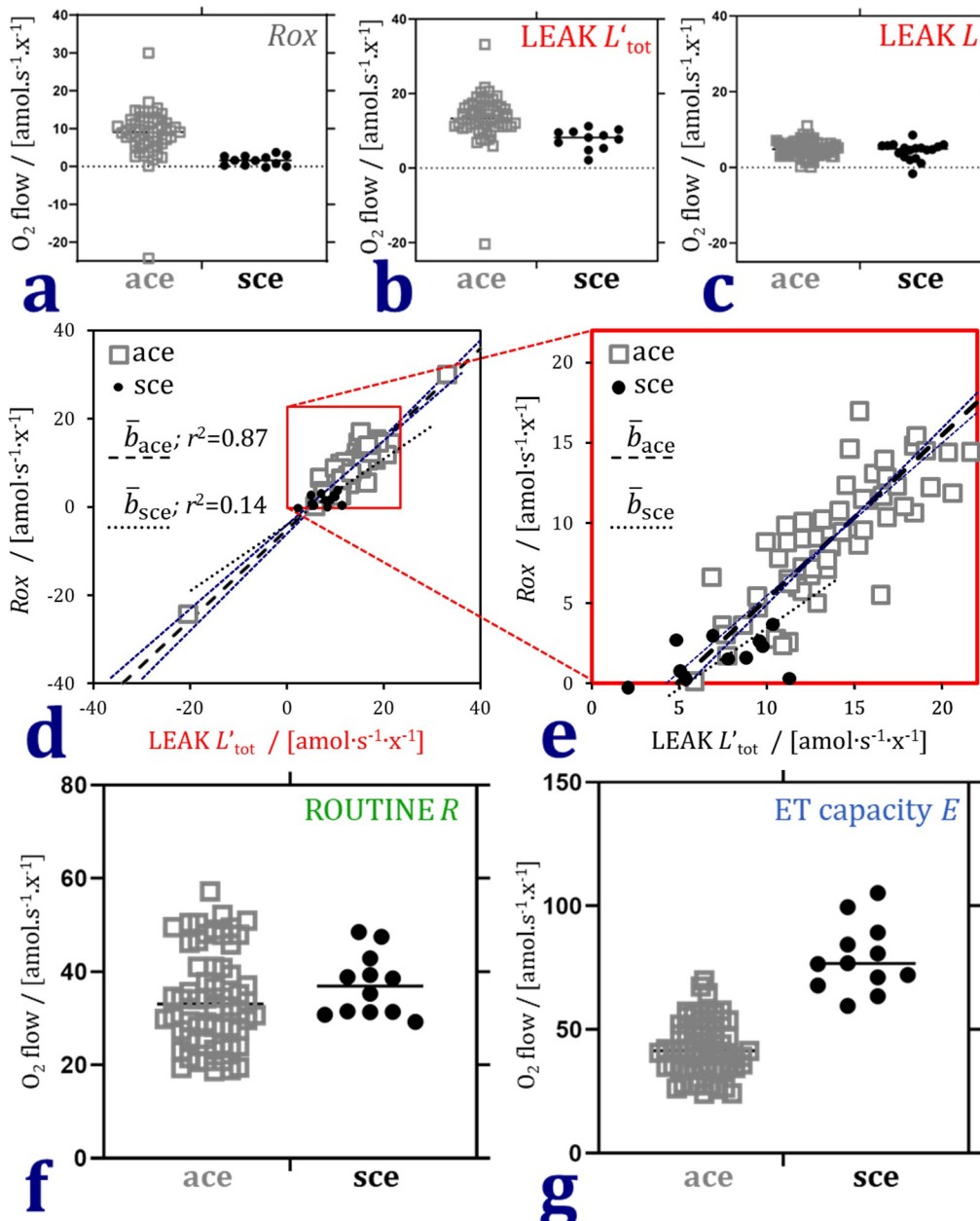

**Fig 3. Residual oxygen consumption *Rox*, LEAK respiration *L*, ROUTINE respiration *R*, and electron transfer capacity *E* in attached and suspended fibroblast cells (ace and sce).** (a) *Rox* measured after inhibition by rotenone and antimycin A reflects instrumental resolution independent of cell physiological conditions for comparison of the two respirometers (ace: XF24, sce: O2k). (b) Total LEAK respiration $L'_{tot}$ uncorrected for *Rox*, higher in ace than sce. (c) *Rox*-correction of LEAK respiration *L* eliminates the difference between ace and sce. (d) Relation between *Rox* and $L'_{tot}$ (full range). $\bar{b}$ is the mean slope of the ordinate and inverted slopes which are shown for ace (thin dashed lines). The low coefficient of determination $r^2$ for sce is related to the very low scatter of the data. (e) Zoom into the positive range of values in panel d. (f) *Rox*-corrected ROUTINE respiration was similar in ace and sce. (g) *Rox*-corrected electron transfer capacity was lower in ace than sce.

## Discussion

Cells cultured attached to a physical substrate undergo many changes when they are detached and maintained in suspension, but possible bioenergetic respiratory changes have not yet been

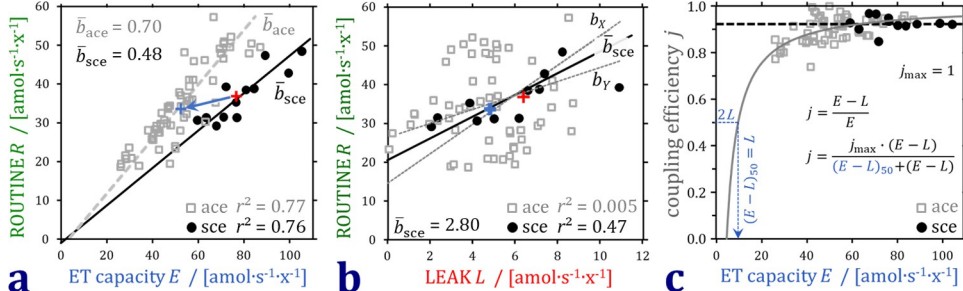

**Fig 4. Bioenergetic cluster analysis of respiration in different respiratory coupling states in attached and suspended cells (ace and sce).** (a) $R$ varies in direct proportion to ET capacity. The linear $R/E$ regressions are different for ace and sce, thus the data points are arranged as separate heterolinear clusters with different $\bar{b}_{ace}$ and $\bar{b}_{sce}$. $E$ is lower in ace than sce at similar $R$ (blue arrow pointing to the left from the median of sce to the median of ace). (b) ace and sce are in the same cluster in the plot of $R$ versus $L$, with high noise in ace LEAK respiration. For sce, $b_Y$ and $b_X$ indicate the slopes for the ordinary and inverted least squares regressions. The trend of $R$ increasing linearly with $L$ and a positive intercept agree with results on different human fibroblasts [25]. (c) A constant coupling control efficiency $j$ at declining $E$ (dotted line for sce) is predicted at constant mitochondrial quality with proportional decline of $E$ and $L$. The relationship between coupling efficiency and $E$ at constant $L$ is hyperbolic (full line fitted for ace). $(E\text{-}L)_{50}$ is the net ET capacity when coupling efficiency is 50%. The maximum coupling efficiency, $j_{max} \cong 1$, is approached with increasing $E$.

addressed. The production of mRNA in fibroblasts is reduced to 20% within a few hours of suspension [10]. The mRNA is not degraded but inactivated and its amount remains constant [10]. The rate of protein synthesis declines more slowly, even slower than expected from the mRNA lifetime of about nine hours [26]. Nevertheless, the suspended cells contain — after almost 3 days of reduced messenger RNA production — unchanged levels of cytoplasmic poly-adenylated RNA due to a stabilization of mRNA against normal turnover. A continuous decline in protein synthesis starts after 12 h in suspension [10]. The recovery of protein synthesis is rapid after reattachment of cells to a tissue culture dish and reactivation of the sequestered mRNA [27]. These responses of cells to suspension or attachment suggest the possibility that altered physical configuration and cell morphology may induce metabolic responses [10]. To our knowledge, the bioenergetic consequences of suspending fibroblasts grown in a mono-layer have not yet been quantitatively assessed, despite numerous studies reporting fibroblast respiration of either suspended or attached cells. Our results showed that ROUTINE respiration was not different in attached cells studied after overnight seeding versus freshly suspended cells measured one hour after harvesting. These findings are relevant for interpreting respiro-metric short-term studies, but do not exclude severe bioenergetic alterations and even cell death as a consequence of prolonged suspension of fibroblasts. Importantly, cells do not immediately change their morphology when re-plated after trypsinization [28].

For a comparison of respiration of attached versus suspended fibroblasts, it was necessary to use two different platforms. Multiwell microchambers contain attached cells without stir-ring. In contrast, the 0.5-mL twin chambers that are stirred continuously to maintain cells in homogenous suspension and to avoid $O_2$ diffusion gradients which would compromise high-resolution respirometry. Our approach covered a time range that is relevant for current tech-niques of measuring respiration in suspended fibroblasts. The duration of maintaining cells in suspension extends from trypsinization to the actual respirometric measurement, which limits the duration to less than two hours from detachment to monitoring of ROUTINE respiration. Importantly, even respiratory measurement of adherent cells in the XF24 did not represent a direct and undisturbed monitoring of oxygen consumption of attached cultured cells but required harvesting of the cells by detachment on the day before measurement. Our study,

however, provides the rationale for extended studies of the stress-response of cells suspended for prolonged periods of time.

Primarily, therefore, we had to evaluate the quantitative agreement between measurements with the two instruments. Higher rates of residual oxygen consumption *Rox* and total LEAK respiration not corrected for *Rox* were obtained in the XF24. This may be attributed to variable instrumental background rather than the attached state of the cells, which is supported by the fact that *Rox*-corrected LEAK respiration was not different between attached and suspended cells (ace and sce). Our results can be compared with a large database on respiration of normal human dermal fibroblasts NHDF measured with the Seahorse XF96 Analyzer and used as controls for the diagnosis of inherited mitochondrial diseases [6]. NHDF data from 2630 wells are summarized in a meta-analysis based on BCA [25]. Respiration was normalized for the count of seeded cells (20 000 cells per well in the XF96, compared to 35 000 cells per well in the XF24 and 375 000 cells in the O2k chamber in the present study). After conversion of the NHDF raw data to SI units, deletion of runs with missing data points, further elimination of 7% of outliers ([6] report 17% outliers), and log transformation to account for positive skewness, the NHDF data are expressed as means ± SD after linear back-transformation. *R*, *L* and *E* are $37 \pm 12$; $6 \pm 3$; and $78 \pm 26$ amol·s$^{-1}$·x$^{-1}$, respectively [25]. This agrees with our results on suspended cells in the O2k and attached cells in the XF24 for *R* and *L* (Table 4). Our data on *E* in the XF24 after 2 μM oligomycin and uncoupler titration up to 0.7 μM were lower ($52 \pm 8$) compared to $78 \pm 25$ in the XF96 at a lower oligomycin concentration (1 μM) and higher FCCP concentration (1 μM; [6]). This supports the interpretation that ET capacity was underestimated in the present study in the XF24 due to an insufficiently high uncoupler concentration (Fig 2A) combined with a high concentration of oligomycin inhibiting *E*.

The variability of respiration normalized for the cell count is of interest from two points of view. (*1*) A methodological perspective: Is high variability mainly the result of respirometric noise, noise in the cell count, or variability introduced by the addition of cells to the respirometric chambers or wells? Is the reproducibility linked to specific techniques and to the magnitude of the cell count used in an assay? (*2*) A physiological perspective: Is the variability factual rather than artefactual [25]? A possible reason for higher variability of residual oxygen consumption obtained in the XF24 is the instrumental background $O_2$ rate, which is not determined in the experimental wells but is measured in parallel only in four separate control wells without cells (Table 1).

Taken together, neither LEAK respiration nor ROUTINE respiration were different when comparing attached and suspended cells. Bioenergetic cluster analysis (BCA) identified the lower coupling control efficiency obtained in the XF24 to be caused by an underestimation of ET capacity as opposed to uncoupling (Fig 4C). Further studies of respiration in different attached and suspended cell types are of great interest in cell physiology, particularly in neuronal and blood cells, and importantly, in cancer cell lines and other cell models of disease. The present approach includes BCA and provides a guideline for extending databases on cell respiration across instrumental platforms, emphasizing the importance of harmonization of protocols.

## Author Contributions

**Conceptualization:** Lucie Zdrazilova, Hana Hansikova, Erich Gnaiger.

**Data curation:** Lucie Zdrazilova.

**Formal analysis:** Lucie Zdrazilova, Erich Gnaiger.

**Funding acquisition:** Hana Hansikova.

**Investigation:** Lucie Zdrazilova, Erich Gnaiger.

**Methodology:** Lucie Zdrazilova, Erich Gnaiger.

**Project administration:** Hana Hansikova.

**Resources:** Hana Hansikova.

**Supervision:** Erich Gnaiger.

**Visualization:** Lucie Zdrazilova, Erich Gnaiger.

**Writing – original draft:** Lucie Zdrazilova, Erich Gnaiger.

**Writing – review & editing:** Hana Hansikova, Erich Gnaiger.

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
