## [Decision Letter · Decision Letter 0]

26 Nov 2021

PONE-D-21-34792Comparable respiratory activity in attached and suspended human fibroblastsPLOS ONE

Dear Dr. Zdrazilova,

Thank you for submitting your manuscript to PLOS ONE. After careful consideration, we feel that it has merit but does not fully meet PLOS ONE’s publication criteria as it currently stands. Therefore, we invite you to submit a revised version of the manuscript that addresses the points raised during the review process.

While the reviewers have underlined the technical interest of your paper in research on mitochondria, it is however clear that the paper would benefit from a more precise writing, alond the lines mentioned by reviewer 3. In particular, the time range during which suspended and attached fibroblats keep similar respiration is a key issue that should be at the very least thoroughly discussed, if not experimentally verified. If you authors have any experiments in store along this line it would be a great idea to include them in a revision.

We look forward to receiving your revised manuscript.

Kind regards,

Thierry Rabilloud

Academic Editor

PLOS ONE

Journal Requirements:

2. Please provide additional details regarding participant consent. In the Methods section, please ensure that you have specified (1) whether consent was informed and (2) what type you obtained (for instance, written or verbal). If your study included minors, state whether you obtained consent from parents or guardians. If the need for consent was waived by the ethics committee, please include this information.

"I have read the journal's policy and the authors of this manuscript have the following competing interests: EG is founder and CEO of Oroboros Instruments, Innsbruck, Austria"

We note that you received funding from a commercial source: "Oroboros Instruments, Innsbruck, Austria"

"This work was partially funded by the European Union’s Horizon 2020 research and innovation programme under grant agreement No. 859770, NextGen-O2k project (EG), Institutional projects GAUK110119 and SVV–UK 260367 (LZ) and by the Ministry of Health of the Czech Republic NV19-07-00149 (HH). Contribution to COST Action CA15203 MitoEAGLE with financial support of Short-Term Scientific missions (LZ)."

"This work was partially funded by the European Union’s Horizon 2020 research and innovation programme under grant agreement No. 859770, NextGen-O2k project (EG),  Institutional projects GAUK110119 and SVV–UK 260367 (LZ) and by theFC49 (HH). Contribution to COST Action CA15203 MitoEAGLE with financial support of Short-Term Scientific missions (LZ)."

Reviewers' comments:

Reviewer's Responses to Questions

**Comments to the Author**

1. Is the manuscript technically sound, and do the data support the conclusions?

Reviewer #1: Yes

Reviewer #2: Yes

Reviewer #3: Partly

2. Has the statistical analysis been performed appropriately and rigorously? 

Reviewer #1: Yes

Reviewer #2: Yes

Reviewer #3: Yes

3. Have the authors made all data underlying the findings in their manuscript fully available?

Reviewer #1: Yes

Reviewer #2: Yes

Reviewer #3: Yes

4. Is the manuscript presented in an intelligible fashion and written in standard English?

Reviewer #1: Yes

Reviewer #2: Yes

Reviewer #3: No

5. Review Comments to the Author

Reviewer #1: The manuscript by Zdrazilova et al. contains a valuable bioenergetic/respirometric comparison of either attached or suspended fibroblasts. This comparison was done in two setups, the attached cells were analyzed in the Seahorse analyzer and the suspended cells in the Oroboros O2k. No differences were observed in ROUTINE respiration of living cells and LEAK respiration obtained after inhibition of ATP synthase by oligomycin. Thus, both approaches may be comparatively or alternatively used.

The authors further find that respiratory values have greater standard deviations in the Seahorse instrumentation. Further, these measurements may also be limited in the number of possible additions, thereby resulting in a potentially underestimated maximal respiratory capacity due to a potential oligomycin overdose or due to a too low dose of uncoupler. On the other hand, the O2k has a limited sample number that can be analyzed in parallel and also requires around ten times more cells per measurement.

For “mitochondriologists”, i.e. bio-medical researchers interested in bioenergetic analyses, this is a valuable paper, as to my knowledge, such direct methodological comparisons hardly exist. Furthermore, the idea to further develop a unifying data handling to allow comparisons of the different approaches is most welcome.

Specific point of criticism/improvement are:

1. With respect to the methodological comparison, a table summarizing the most important differences (e.g. possible sample replicates, required cell amount, number of possible additions, detection mode, sample volume, etc…) between the approaches would be very helpful for the reader to decide which approach to use.

2. I doubt that the unifying respiratory capacity per single cell unit is a good one. First, the resulting attomol values are not really of practical use. Second, cell research typically occurs on “millions”. Thus, such a more practical number could be better/alternatively introduced. Third, when looking at the standard deviations of 30% or more (see the mentioning of R, L and E that are 37 ± 12; 6 ± 3; and 78 ± 26 amol∙s1∙x1 in reference 18), one may think of alternative/more precise parameters?

3. The authors state: “Our results suggest that ROUTINE respiration was not different in attached cells studied after overnight seeding and freshly suspended cells measured one hour after harvesting.” This comes with an important question: how long can detached solubilized cells be “safely” measured until clear respiratory deficits with respect to attached cells can be encountered? It would be wonderful if the authors could do such measurements at different time points to provide data for this. Clearly, such an information would further increase the value of this manuscript for the interested readers.

Minors:

1. At different passages in the text the authors speak about “physiological or near-physiological conditions”. In which sense? For example, the oxygen concentration in the measurement chambers is assumed to be much higher than in tissue? Thus, please specify.

2. One/two sentence/s explaining why fibroblasts were chosen for this study would be appreciated.

3. The authors find that ROX and LEAK measurements were higher in Seahorse than in the O2k and with a higher scatter. Could this be due to the detection via fluorescence that may be non-linear in this low range of change? Please comment why this could be in the discussion.

Reviewer #2: The study by Zdrazilova and co-workers addresses an important point regarding the assessment of the mitochondria-driven cell respiration in intact cells. The recent introduction of methodologies for measuring O2 consumptions, alternative to the traditional Clark-electrode-based polarography, have raised the possibilities of differences in determining the absolute oxygen consumption rates. In addition to a complete different instrumental design, the Seahorse approach carries measurements in attached cells while the electrodic measurements are conducted in suspended cells. This raises the reservation of a different bioenergetics response of the cell samples depending if they are anchored (better mimicking a physiological situation) or are freely floating in suspension. Zdrazilova and co-workers uncover this gap of knowledge comparing the respirometric parameters of human fibroblasts attained by real-time multiwall and high resolution oximetry. The results reported do not unveil significant differences between adherent and short-term suspended cells in the so called Routine and Leak respirations (after correction for mitochondria-independent respiration) but a slower oxygen consumption rate (i.e. the maximal capacity) in suspended cells that the authors explain as due to an inhibitory effect of oligomycin used at higher concentrations in the Seahorse standard protocol.

The study is well-conducted, the results are clearly presented in the Figures and table and supported by a robust statistical analysis. The recommendations presented in the supplements are also relevant and helpful in providing a standardized protocol to set reproducibility in this specific assay to extend respirometric database.

Only the following minor points are asked to replay:

1. Can the authors anticipate, if they have preliminary evidence, that the same conclusions attained in this study using primary cell cultures might be extended to cancer cell lines that are the samples more widely used in cellular biochemistry investigations?

2. The oxygen consumption rate under uncoupled condition is taken as a measure of the maximal capacity of the respiratory chain that is untied by the controlling protonmotive force. However, anionic respiratory substrates (such as pyruvate or glutamate) enter into the mitochondria by transporters utilizing the mt-DpH gradient. Therefore, under conditions dissipating the delta-mu-H+ the respiratory capacity is, in any case, limited by the respiratory substrate availability. It is also possible that under this condition the cell switches to substrates whose entry into mitochondria is not delta-mu-H+-driven (such as fatty acids). Can the authors comment on this and suggest an alternative term in place of the somewhat confusing “maximal capacity”?

Reviewer #3: In this study, Lucie Zdrazilova and coworkers have compared the oxygen consumption of human dermal fibroblast cell lines cultured in monolayers, either directly in their adherent form, or in suspension after trypsinization. To address this question, they used two instruments, the Seahorse XF Analyser (Agilent, US) which is designed to measure respiration on adherent cells and the Oroboros O2k (Oroboros Instruments, Austria) which is designed for suspended cells. Their main conclusion is that short-term suspension of fibroblasts does not affect respiratory activity and coupling control.

This article needs some proofreading to improve the English. As an example, the last sentence of the abstract is particularly muddled: “Consistent results obtained with different platforms provide a test for reproducibility and allow for building an extended respirometric database” which could be rewritten as “Obtaining consistent results across different platforms is a good measure of reproducibility and could help build a comprehensive respirometric database.”

The introduction and the discussion sections are insufficiently developed. The introduction is particularly short. In contrast, the materials and methods and results sections are better developed and provide sufficient information. This paper is methodological and the study is centered on the use of the new Seahorse apparatus. However the experiments are in my opinion not very original and do not present an important breakthrough. While the results do not show significant differences between measurements in adherent or in suspended conditions, the authors suggest that it could probably be different after longer incubation times in suspension, but show no experiments to support this hypothesis.

The introduction lacks a clear explanation of the biological interest and the real goal of this study. The bibliographic references of the introduction are particularly old and are not cited with sufficient relevance. For example, at the beginning of the introduction “After blebbing, cells undergo membrane reorganization and attain a spherical shape to prevent membrane loss [3].” This phrase alone, taken out of context, is very hard to fully understand and in my opinion does not accurately convey the message of reference 3.

The works cited in the discussion are also not clearly exposed.

At the beginning of the discussion, for example: “Suspending fibroblasts causes an immediate drop of mRNA synthesis to about 20 % of controls within minutes [6]. The remaining mRNA is inactivated, such that the total cell mRNA content remains unchanged.” As I did not quite grasp the meaning of these sentences, I read the cited article. My impression is that the authors have mixed up two separate concepts. Indeed, the cited article states that the mRNAs were in fact stabilized so that the total amount of mRNA remained constant, and that they were also probably transiently inactivated to explain the drop in protein synthesis and the rapid increase of protein synthesis which occured upon reattachment to a solid substrate (prior to mRNA neosynthesis). But it is incorrect to say that the inactivation of the remaining mRNA explains “that the total cell mRNA content remains unchanged”.

The discussion does not give a strong argument to justify the technical and or biological interest of this study. However, this study could probably be helpful to address technical questions about the Seahorse instrument. For example, does this study prove that trypsinization induces no bias when measuring respiration on suspended cells?

6. PLOS authors have the option to publish the peer review history of their article (what does this mean?). If published, this will include your full peer review and any attached files.

Reviewer #1: No

Reviewer #2: No

Reviewer #3: No

---

## [Author Response · Author response to Decision Letter 0]

10 Jan 2022

Response to reviewers (Manuscript PONE-D-21-34792)

We have carefully read reviewers comments (indicated in italics) and made several changes in according to their suggestions. Our response to a reviewers comments are in bold and changes made in manuscript are described in red. 

Reviewers' comments:

Reviewer #1: “The manuscript by Zdrazilova et al. contains a valuable bioenergetic/respirometric comparison of either attached or suspended fibroblasts. This comparison was done in two setups, the attached cells were analyzed in the Seahorse analyzer and the suspended cells in the Oroboros O2k. No differences were observed in ROUTINE respiration of living cells and LEAK respiration obtained after inhibition of ATP synthase by oligomycin. Thus, both approaches may be comparatively or alternatively used.

The authors further find that respiratory values have greater standard deviations in the Seahorse instrumentation. Further, these measurements may also be limited in the number of possible additions, thereby resulting in a potentially underestimated maximal respiratory capacity due to a potential oligomycin overdose or due to a too low dose of uncoupler. On the other hand, the O2k has a limited sample number that can be analyzed in parallel and also requires around ten times more cells per measurement.

For “mitochondriologists”, i.e. bio-medical researchers interested in bioenergetic analyses, this is a valuable paper, as to my knowledge, such direct methodological comparisons hardly exist. Furthermore, the idea to further develop a unifying data handling to allow comparisons of the different approaches is most welcome.”

Specific point of criticism/improvement are:

1. “With respect to the methodological comparison, a table summarizing the most important differences (e.g. possible sample replicates, required cell amount, number of possible additions, detection mode, sample volume, etc…) between the approaches would be very helpful for the reader to decide which approach to use.”

We agree with this suggestion and included Table 1 to the main text (page 4 line 120).

Manuscript:

Table 1. Comparative specifications of Seahorse XF24 and Oroboros O2k applied in the present coupling control protocol.

 Seahorse XF24 Oroboros O2k

wells or chambers per instrument 24 semiclosed wells 2 diffusion-tight chambers

instrumental background and experiments with cells 4 separated wells without cells and 20 wells with cells in parallel two identical chambers serially without and with cells

detection mode optical O2 sensor with fluorophores electrochemical polarographic oxygen sensor

temperature control 37 °C (stability ±0.1°C in a tray) 37 °C (stability ±0.002 °C in a copper block)

limit of detection of oxygen flux not specified ±1 pmol O2∙s-1∙mL-1

cell conditions attached cells cell suspension

required cell amount 35 000 375 000

sample volume [µL] 450 540

closed chamber volume 7 µL when sensor probes are inserted 500 µL when stoppers are inserted

titrations automatic manual

number of possible titrations 4 technically unlimited

experimental duration [min] 120 30 to 40

2. “I doubt that the unifying respiratory capacity per single cell unit is a good one. First, the resulting attomol values are not really of practical use. Second, cell research typically occurs on “millions”. Thus, such a more practical number could be better/alternatively introduced. Third, when looking at the standard deviations of 30% or more (see the mentioning of R, L and E that are 37 ± 12; 6 ± 3; and 78 ± 26 amol∙s1∙x1 in reference 18), one may think of alternative/more precise parameters?”

The unit ‘mol∙s-1∙x-1 has been recommended by a consortium of 666 coauthors, elaborated in the framework of a European Union funded COST Action MitoEAGLE, and published last year; 


https://www.bioenergetics-communications.org/index.php/bec/article/view/gnaiger_2020_mitophysiology

For further clarification, on page 7 line 234 of the original MS, we provide the link from this unit to the equivalent expression per million cells: “[amol∙s-1∙x-1] equivalent to [pmol∙s-1∙(106 x)-1].”

“Third, when looking at the standard deviations of 30% or more (see the mentioning of R, L and E that are 37 ± 12; 6 ± 3; and 78 ± 26 amol∙s1∙x1 in reference 18), one may think of alternative/more precise parameters?”

We added the following text on page 12, line 433:

Manuscript: The variability of respiration normalized for the cell count is of interest from two points of view. (1) A methodological perspective: Is high variability mainly the result of respirometric noise, noise in the cell count, or variability introduced by the addition of cells to the respirometric chambers or wells? Is the reproducibility linked to specific techniques and to the magnitude of the cell count used in an assay? (2) A physiological perspective: Is the variability factual rather than artefactual [25]? 

3. “The authors state: “Our results suggest that ROUTINE respiration was not different in attached cells studied after overnight seeding and freshly suspended cells measured one hour after harvesting.” This comes with an important question: how long can detached solubilized cells be “safely” measured until clear respiratory deficits with respect to attached cells can be encountered? It would be wonderful if the authors could do such measurements at different time points to provide data for this. Clearly, such an information would further increase the value of this manuscript for the interested readers.”

We added some of the sentences into the discussion part, please see below (page 11 line 398):

 Manuscript: Our approach covered a time range that is relevant for current techniques of measuring respiration in suspended fibroblasts. The duration of maintaining cells in suspension extends from trypsinization to the actual respirometric measurement, which limits the duration to less than two hours from detachment to monitoring of ROUTINE respiration. Importantly, even respiratory measurement of adherent cells in the XF24 did not represent a direct and undisturbed monitoring of oxygen consumption of attached cultured cells but required harvesting of the cells by detachment on the day before measurement. Our study, however, provides the rationale for extended studies of the stress-response of cells suspended for prolonged periods of time.

Minors:

1. “At different passages in the text the authors speak about “physiological or near-physiological conditions”. In which sense? For example, the oxygen concentration in the measurement chambers is assumed to be much higher than in tissue? Thus, please specify.”

Experimental oxygen concentrations near air saturation are effectively hyperoxic relative to oxygen levels within tissues. Respiratory function, however, is independent of oxygen concentration from air to tissue oxygen levels. Nevertheless, we agree that near-physiological conditions of oxygenation have to be considered when studying different critical functions (page 6 line 177).

Manuscript: Experimental oxygen concentrations near air saturation (190 µM to 140 µM; Fig 1b and d) are much higher than extracellular oxygen concentrations in various tissues in vivo. Respiration of living cells, however, is independent of oxygen pressure from air saturation to tissue-level oxygen concentration, the latter ranging between 50 and 10 µM [19]. Therefore, even if high experimental oxygen concentrations are not physiological, respiratory measurements are not affected. This is in direct contrast to cellular hydrogen peroxide production, which increases with oxygen concentration over the entire experimental oxygen concentration range [20].

2. “One/two sentence/s explaining why fibroblasts were chosen for this study would be appreciated.”

According to reviewers comment, we added some text to the introduction (page 3 line 49)

Manuscript: Fibroblast cell lines are established models routinely applied in studies of mitochondrial diseases [6; 7; 8; 9]. These cells can be investigated in culture either attached to the surface of an experimental chamber or in suspension after detachment.

3. “The authors find that ROX and LEAK measurements were higher in Seahorse than in the O2k and with a higher scatter. Could this be due to the detection via fluorescence that may be non-linear in this low range of change? Please comment why this could be in the discussion.”

LEAK respiration was not higher in the XF24 after baseline correction for Rox (Table 4). We added a paragraph to the discussion session (page 12 line 438). 

Manuscript: A possible reason for higher variability of residual oxygen consumption obtained in the XF24 is the instrumental background O2 rate, which is not determined in the experimental wells but is measured in parallel only in four separate control wells without cells (Table 1).

Reviewer #2: “The study by Zdrazilova and co-workers addresses an important point regarding the assessment of the mitochondria-driven cell respiration in intact cells. The recent introduction of methodologies for measuring O2 consumptions, alternative to the traditional Clark-electrode-based polarography, have raised the possibilities of differences in determining the absolute oxygen consumption rates. In addition to a complete different instrumental design, the Seahorse approach carries measurements in attached cells while the electrodic measurements are conducted in suspended cells. This raises the reservation of a different bioenergetics response of the cell samples depending if they are anchored (better mimicking a physiological situation) or are freely floating in suspension. Zdrazilova and co-workers uncover this gap of knowledge comparing the respirometric parameters of human fibroblasts attained by real-time multiwall and high resolution oximetry. The results reported do not unveil significant differences between adherent and short-term suspended cells in the so called Routine and Leak respirations (after correction for mitochondria-independent respiration) but a slower oxygen consumption rate (i.e. the maximal capacity) in suspended cells that the authors explain as due to an inhibitory effect of oligomycin used at higher concentrations in the Seahorse standard protocol.

The study is well-conducted, the results are clearly presented in the Figures and table and supported by a robust statistical analysis. The recommendations presented in the supplements are also relevant and helpful in providing a standardized protocol to set reproducibility in this specific assay to extend respirometric database. Only the following minor points are asked to replay:”

Following this comment, we moved the supplementary tables to the methods section. (Tables 2 and 3).

1. “Can the authors anticipate, if they have preliminary evidence, that the same conclusions attained in this study using primary cell cultures might be extended to cancer cell lines that are the samples more widely used in cellular biochemistry investigations?”

We agree that cancer cells are abundantly studied by presented two instruments (as an example from many manuscripts: Cochrane et al, 2021; Aguilar-Valdés et al, 2021) and therefore we believe that comparing their respiration in suspended and attached state is an important approach. Nevertheless, such a comparison using cancer cells hasn’t been studied yet and we can only hypothesize about possible results.

Cochrane EJ, Hulit J, Lagasse FP, Lechertier T, Stevenson B, Tudor C, Trebicka D, Sparey T, Ratcliffe AJ. Impact of Mitochondrial Targeting Antibiotics on Mitochondrial Function and Proliferation of Cancer Cells. ACS Med Chem Lett. 2021 Mar 8;12(4):579-584. doi: 10.1021/acsmedchemlett.0c00632. 

Aguilar-Valdés A, Noriega LG, Tovar AR, Ibarra-Sánchez MJ, Sosa-Hernández VA, Maravillas-Montero JL, Martínez-Aguilar J. SWATH-MS proteomics of PANC-1 and MIA PaCa-2 pancreatic cancer cells allows identification of drug targets alternative to MEK and PI3K inhibition. Biochem Biophys Res Commun.2021 May 7;552:23-29. doi: 10.1016/j.bbrc.2021.03.018. 

 In line 448 of the original MS, we add: 

Manuscript: Further studies of respiration in different attached and suspended cell types are of great interest in cell physiology, particularly in neuronal and blood cells, and importantly, in cancer cell lines and other cell models of disease.

2. “The oxygen consumption rate under uncoupled condition is taken as a measure of the maximal capacity of the respiratory chain that is untied by the controlling protonmotive force. However, anionic respiratory substrates (such as pyruvate or glutamate) enter into the mitochondria by transporters utilizing the mt-DpH gradient. Therefore, under conditions dissipating the delta-mu-H+ the respiratory capacity is, in any case, limited by the respiratory substrate availability. It is also possible that under this condition the cell switches to substrates whose entry into mitochondria is not delta-mu-H+-driven (such as fatty acids). Can the authors comment on this and suggest an alternative term in place of the somewhat confusing “maximal capacity”?”

We agree that the term “maximal capacity” may be confusing, and this is why we don’t use it. We clearly define and use the term “maximum electron transfer capacity” in the context of multiple uncoupler titrations, or “maximum rate of O2 consumption reflecting the electron transfer capacity E”. We compare this term published in a MitoEAGLE consensus paper with the term ‘maximal respiration’ widely used in conjunction with the XF24 or XF96 (Table 2, moved from supplemental table S1). Instead of “maximal capacity” we suggest using the term “electron transfer capacity” or more explicitly “Rox-corrected electron transfer capacity”. Addressing the question on the pmF dependence of substrate transport into the mt-matrix is relevant, but requires analytical studies with mitochondrial preparations rather than living cells. 

Reviewer #3: “In this study, Lucie Zdrazilova and coworkers have compared the oxygen consumption of human dermal fibroblast cell lines cultured in monolayers, either directly in their adherent form, or in suspension after trypsinization. To address this question, they used two instruments, the Seahorse XF Analyser (Agilent, US) which is designed to measure respiration on adherent cells and the Oroboros O2k (Oroboros Instruments, Austria) which is designed for suspended cells. Their main conclusion is that short-term suspension of fibroblasts does not affect respiratory activity and coupling control.

This article needs some proofreading to improve the English. As an example, the last sentence of the abstract is particularly muddled: “Consistent results obtained with different platforms provide a test for reproducibility and allow for building an extended respirometric database” which could be rewritten as “Obtaining consistent results across different platforms is a good measure of reproducibility and could help build a comprehensive respirometric database.”

We appreciate the effort of the reviewer to point out the potential of improving the phrasing in several instances to help us give a clear message to the reader. We took reviewer’s advice and sent the manuscript to English revision. We agree that the last sentence in the abstract needs to be edited in an improved version. We modified the sentence (page 2 line 19): 

Manuscript: Evaluation of results obtained by different platforms provides a test for reproducibility beyond repeatability. Repeatability and reproducibility are required for building a validated respirometric database.

“The introduction and the discussion sections are insufficiently developed. The introduction is particularly short. In contrast, the materials and methods and results sections are better developed and provide sufficient information. This paper is methodological and the study is centered on the use of the new Seahorse apparatus. However the experiments are in my opinion not very original and do not present an important breakthrough. While the results do not show significant differences between measurements in adherent or in suspended conditions, the authors suggest that it could probably be different after longer incubation times in suspension, but show no experiments to support this hypothesis.”

We meant to state it as a cautionary note, that extrapolation of the present results to longer incubation times is not justified without specific testing. We included such a clarification to the text with more precise discussion (page 11 line 398). We refer to our response to the related comment by reviewer 1.

 Manuscript: Our approach covered a time range that is relevant for current techniques of measuring respiration in suspended fibroblasts. The duration of maintaining cells in suspension extends from trypsinization to the actual respirometric measurement, which limits the duration to less than two hours from detachment to monitoring of ROUTINE respiration. Importantly, even respiratory measurement of adherent cells in the XF24 did not represent a direct and undisturbed monitoring of oxygen consumption of attached cultured cells but required harvesting of the cells by detachment on the day before measurement. Our study, however, provides the rationale for extended studies of the stress-response of cells suspended for prolonged periods of time.

“The introduction lacks a clear explanation of the biological interest and the real goal of this study. “

We modified the introductory paragraph (page 3 line 67).

Manuscript: The first aim of the present study was the evaluation of respiration in attached compared to suspended fibroblasts. The Seahorse XF Analyzer (Agilent, US) is designed for studying respiration of attached cells (ace), whereas the Oroboros O2k (Oroboros Instruments, Austria) is optimized for high-resolution respirometry with suspended cells (sce). Therefore, a platform comparison was required as the second aim of our study. We applied comparable and standardized protocols for monitoring respiratory activity in living cells under near-physiological conditions. 

Adding the new Table 1 (platform comparison) and moving the two supplementary tables into the methods section emphasizes the second aim even more explicitly. 

“The bibliographic references of the introduction are particularly old and are not cited with sufficient relevance. For example, at the beginning of the introduction “After blebbing, cells undergo membrane reorganization and attain a spherical shape to prevent membrane loss [3].” This phrase alone, taken out of context, is very hard to fully understand and in my opinion does not accurately convey the message of reference 3.”

We admit that the used reference 3, is older one, but we believe that it is a very important reference for our paper considering changes in cell morphology after trypsinization. We changed the text (page 3 line 54): 

Manuscript: After trypsinization, fibroblasts undergo membrane reorganization and attain a spherical shape with a so-called blebbed surface morphology to prevent membrane loss by providing transient membrane storage [3].

“The works cited in the discussion are also not clearly exposed. At the beginning of the discussion, for example: “Suspending fibroblasts causes an immediate drop of mRNA synthesis to about 20 % of controls within minutes [6]. The remaining mRNA is inactivated, such that the total cell mRNA content remains unchanged.” As I did not quite grasp the meaning of these sentences, I read the cited article. My impression is that the authors have mixed up two separate concepts. Indeed, the cited article states that the mRNAs were in fact stabilized so that the total amount of mRNA remained constant, and that they were also probably transiently inactivated to explain the drop in protein synthesis and the rapid increase of protein synthesis which occured upon reattachment to a solid substrate (prior to mRNA neosynthesis). But it is incorrect to say that the inactivation of the remaining mRNA explains “that the total cell mRNA content remains unchanged”.

In our text there is no indication that the inactivation of the remaining mRNA explains “that the total cell mRNA content remains unchanged. However, we corrected another mistake(page 11 line 373).

Manuscript: The production of mRNA in fibroblasts is reduced to 20 % within a few hours of suspension [10]. The mRNA is not degraded but inactivated and its amount remains constant [10]. 

“The discussion does not give a strong argument to justify the technical and or biological interest of this study. However, this study could probably be helpful to address technical questions about the Seahorse instrument. For example, does this study prove that trypsinization induces no bias when measuring respiration on suspended cells?”

The reviewer considers that one approach (XF24) studies respiration of adherent cells “directly in their adherent form”. It has to be emphasized even more clearly, that this is not the case. In line 146 of the original MS, we explain: “.The day before measurement, cells were harvested after trypsinization (added in the revision), resuspended in DMEM culture medium, counted, and 35 000 cells were seeded on 20 wells of a 24-well plate for over-night incubation.” The term ‘directly’ implies a different meaning. This is directly relevant to the reviewer’s question: “does this study prove that trypsinization induces no bias when measuring respiration on suspended cells?” Trypsinization is implicated in both approaches. We implemented sentence into the discussion (page 12 line 403)

Manuscript: Importantly, even respiratory measurement of adherent cells in the XF24 did not represent a direct and undisturbed monitoring of oxygen consumption of attached cultured cells but required harvesting of the cells by detachment on the day before measurement.

---

## [Decision Letter · Decision Letter 1]

14 Feb 2022

Comparable respiratory activity in attached and suspended human fibroblasts

PONE-D-21-34792R1

Dear Dr. Zdrazilova,

We’re pleased to inform you that your manuscript has been judged scientifically suitable for publication and will be formally accepted for publication once it meets all outstanding technical requirements.

Kind regards,

Thierry Rabilloud

Academic Editor

PLOS ONE

Additional Editor Comments (optional):

Reviewers' comments:

Reviewer's Responses to Questions

**Comments to the Author**

1. If the authors have adequately addressed your comments raised in a previous round of review and you feel that this manuscript is now acceptable for publication, you may indicate that here to bypass the “Comments to the Author” section, enter your conflict of interest statement in the “Confidential to Editor” section, and submit your "Accept" recommendation.

Reviewer #1: All comments have been addressed

Reviewer #2: All comments have been addressed

Reviewer #3: All comments have been addressed

2. Is the manuscript technically sound, and do the data support the conclusions?

Reviewer #1: Yes

Reviewer #2: Yes

Reviewer #3: Yes

3. Has the statistical analysis been performed appropriately and rigorously? 

Reviewer #1: Yes

Reviewer #2: Yes

Reviewer #3: I Don't Know

4. Have the authors made all data underlying the findings in their manuscript fully available?

Reviewer #1: Yes

Reviewer #2: Yes

Reviewer #3: Yes

5. Is the manuscript presented in an intelligible fashion and written in standard English?

Reviewer #1: Yes

Reviewer #2: Yes

Reviewer #3: Yes

6. Review Comments to the Author

Reviewer #1: (No Response)

Reviewer #2: The authors have sufficiently satisfied the requests of this reviewer although the point relating to the different utilization of respiratory substrates by mitochondria in the presence of a decoupler has been evaded.

Reviewer #3: I appreciate the efforts of the authors to answer to my questions and to improve their article. The new version is good for publication. The authors are acknowledged for their reply to the issues raised.

7. PLOS authors have the option to publish the peer review history of their article (what does this mean?). If published, this will include your full peer review and any attached files.

Reviewer #1: No

Reviewer #2: **Yes: **Nazzareno Capitanio

Reviewer #3: No

---

## [Editor Report · Acceptance letter]

23 Feb 2022

PONE-D-21-34792R1 

Comparable respiratory activity in attached and suspended human fibroblasts 

Dear Dr. Zdrazilova:

I'm pleased to inform you that your manuscript has been deemed suitable for publication in PLOS ONE. Congratulations! Your manuscript is now with our production department. 

Kind regards, 

on behalf of

Dr. Thierry Rabilloud 

Academic Editor

PLOS ONE